# *Ligilactobacillus murinus* Strains Isolated from Mice Intestinal Tract: Molecular Characterization and Antagonistic Activity against Food-Borne Pathogens

**DOI:** 10.3390/microorganisms11040942

**Published:** 2023-04-04

**Authors:** Ivonne Lizeth Sandoval-Mosqueda, Adriana Llorente-Bousquets, Carlos Soto, Crisóforo Mercado Márquez, Silvina Fadda, Juan Carlos Del Río García

**Affiliations:** 1Posgrado, Facultad de Estudios Superiores Cuautitlán, Universidad Nacional Autónoma de Mexico, km 2.5 Carretera Cuautitlán-Teoloyucan, San Sebastian Xhala, Cuautitlán Izcalli 54714, Mexico; 2Ingeniería y Tecnología, Facultad de Estudios Superiores Cuautitlán, Universidad Nacional Autónoma de México, km 2.5 Carretera Cuautitlán-Teoloyucan, San Sebastian Xhala, Cuautitlán Izcalli 54714, Mexico; 3Ciencias Biológicas, Facultad de Estudios Superiores Cuautitlán, Universidad Nacional Autónoma de México, km 2.5 Carretera Cuautitlán-Teoloyucan, San Sebastian Xhala, Cuautitlán Izcalli 54714, Mexico; 4Centro de Referencia para Lactobacilos, Batalla de Chacabuco 145 sur, San Miguel de Tucumán T4000, Argentina

**Keywords:** lactic acid bacteria, *Lactobacillus*, bioinformatics, food-borne pathogens, lytic activity

## Abstract

Considering the objectives of “One Health” and the Sustainable development Goals “Good health and well-being” for the development of effective strategies to apply against bacterial resistance, food safety dangers, and zoonosis risks, this project explored the isolation and identification of *Lactobacillus* strains from the intestinal tract of recently weaned mice; as well as the assessment of antibacterial activity against clinical and zoonotic pathogens. For molecular identification, 16S rRNA gene-specific primers were used and, via BLAST-NCBI, 16 *Ligilactobacillus murinus*, one *Ligilactobacillus animalis,* and one *Streptococcus salivarius* strains were identified and registered in GenBank after the confirmation of their identity percentage and the phylogenetic analysis of the 16 *Ligilactobacillus murinus* strains and their association with *Ligilactobacillus animalis*. The 18 isolated strains showed antibacterial activity during agar diffusion tests against *Listeria monocytogenes* ATCC 15313, enteropathogenic *Escherichia coli* O103, and *Campylobacter jejuni* ATCC 49943. Electrophoretic and zymographic techniques confirmed the presence of bacteriolytic bands with a relative molecular mass of 107 kDa and another of 24 kDa in *Ligilactobacillus murinus* strains. UPLC-MS analysis allowed the identification of a 107 kDa lytic protein as an N-acetylmuramoyl-L-amidase involved in cytolysis and considered a bacteriolytic enzyme with antimicrobial activity. The 24 kDa band displayed similarity with a portion of protein with aminopeptidase function. It is expected that these findings will impact the search for new strains and their metabolites with antibacterial activity as an alternative strategy to inhibit pathogens associated with major health risks that help your solution.

## 1. Introduction

Zoonotic diseases, food safety dangers, and bacterial resistance are related and affect human, animal, and environmental health. Hence, the “One Health” comprehensive approach acknowledges the significant need to develop effective strategies to solve these global health issues [1]. In that sense, food-borne diseases, especially zoonotic diseases, are the cause of morbidity and mortality globally. In accordance with the European Food Safety Authority, *Campylobacter* spp. is the main zoonotic agent that causes diarrheic disease, with 120,946 cases, 8605 hospital admissions, and 45 deaths, which represents over 60% of the cases reported as of 2020 in the European Union. Enteropathogenic *E. coli* is acknowledged as the most important cause of acute diarrheal disease in children under four years old, and Shiga toxin-producing strains of *E. coli* (STEC) are highly virulent and associated with 4446 cases, 652 hospital admissions, and 13 deaths. In this case, enteropathogenic *E. coli* isolated from rabbit kidney (REPEC) O103 was used as the reference strain in this study [2]. *Listeria monocytogenes* is one of the most severe food-borne pathogens, with 1876 cases, 780 hospital admissions, and 167 deaths, with a fatality rate of 30%. The number of cases related to *L. monocytogenes* has increased over the last four years [3].

The use of antimicrobials at a large scale has been documented in animal production. In addition to causing resistant bacteria that, in turn, provoke infections in humans, it has been found that animals excrete a significant percentage (75–90%) of non-metabolized antimicrobials, which are scattered in the environment [4]. The increased resistance to antibiotics in these pathogens contributes to the severity of the diseases they provoke, as observed recently in human and animal isolations of multi-resistant *Campylobacter* spp. and *E. coli* strains [5,6].

Among the key strategies to address the constant infectious illnesses and emerging challenges, the One Health approach and the objective to achieve good health and well-being are considered. Both are essential multi-faceted frameworks to promote good health for everyone. Taking into account the importance of health for sustainable development, the 2030 Agenda highlights the complexity and interconnection of both schemes and considers the emerging global health priorities as well, which are not explicitly included in the sustainable development goals but require action, as it is the case of antimicrobial resistance [7]. Among the development of alternatives to reduce the use of antimicrobials is the use of vaccines, phage therapy and probiotics [4]. Probiotics are defined as “alive microorganisms that, when administered in adequate quantities, have a beneficial impact on the health of the host [8]. Among the studied beneficial effects are the control of pathogens, immunologic effects such as the preservation of the intestinal barrier, and immunomodulatory activity. Lactic Acid Bacteria (LAB) are the primary microorganisms used as probiotics. These are administered with the purpose of controlling pathogen microorganisms and improving the natural defense mechanisms of humans and animals. It has been suggested that the use of these strains in animal production can reduce the risk of pathogen transference from food to humans, and it can be an alternative to the use of antibiotic growth promoters, therefore reducing the development of bacterial resistance [9].

LABs are a heterogeneous group of microorganisms characterized by the production of lactic acid from the fermentation of carbohydrates. Throughout history, they have been used in the production and preservation of fermented food, and in 1988 the Code of Federal Regulations defined them as microorganisms Generally Recognized as Safe (GRAS) [10,11]. The preservation of food with LAB is attributed to the production of several metabolites with antimicrobial activity, such as organic acids, hydrogen peroxide, reuterin, diacetyl, and protein nature substances such as bacteriocins, peptidoglycan hydrolases (PGH) or bacteriocin-like inhibitory substances (BLIS) [12,13]. Several studies have shown that LAB and their metabolites can inhibit pathogen microorganisms, such as *L. monocytogenes*, *Staphylococcus aureus*, *Salmonella enterica*, and *E. coli*, so they represent an important line of research among different groups that look for the characterization of bacterial antagonism phenomena as an alternative against resistant pathogens for their use in animal production, safe food, and medicine [9,14].

LAB have been isolated from a variety of niches, generally plants and food [15], from human and animal microbiota [16,17], as well as from soil [18] and marine sediments [19]. The animal gastrointestinal tract represents the source with the highest potential to obtain LAB [16], mainly *Lactobacillus* spp. Specifically, huge proportions of *Lactobacillus* have been identified in experimental mice, as well as the importance of this genera in the control of pathogens and the prevention of gut dysbiosis [20,21]. Therefore, in this work, the objective was the isolation of *Lactobacillus* strains from mice intestinal tract, their identification at molecular and phylogenetic levels, as well as the evaluation of their in vitro antibacterial activity against the pathogens *L. monocytogenes*, enteropathogenic *E. coli* and *C. jejuni*.

It is expected that the results of this study will impact the detection of new strains and their metabolites with antibacterial activity in the search for strategies for the inhibition of these pathogens of clinical and zoonotic interest. Furthermore, the presence of enzymes with broad-spectrum bacteriolytic activity can be safely applied as promoters of animal health growth as an alternative to the use of antibiotics for the control of antibiotic-resistant pathogenic bacteria.

## 2. Materials and Methods

### 2.1. Isolation and Selection of LAB

LAB isolation was performed from just-weaned CD1 mice collected from the Animal Facility, Multidisciplinary Research Unit in FES Cuautitlan, UNAM. The project was evaluated and approved by the CICUAE—Institutional Committee for the Care and Use of Experimental Animals with protocol number C18_20, and the euthanasia was performed in accordance with NOM-062-ZOO-1999 [22]. Under sterile conditions, the duodenum, jejunum, and colon were extracted from each mouse in order to obtain the intestinal contents by washing through the lumen of each intestine portion with sterile peptone water (NaCl 8.5 mg·mL^−1^, proteose peptone 1 mg·mL^−1^). The collected intestinal matter was inoculated in Lactobacilli MRS broth (De Man Rogosa and Sharp, BD Difco^TM^, USA), in proportion (1:10) and incubated at 30 ± 2 °C for 24 h in a jar anaerobic system (BD GasPak^TM^, USA). Subsequently, streak seeding was performed in MRS agar incubation at 30 ± 2 °C for 48 h per intestine portion. The round, white colonies of 1–2 mm that, when stained, were Gram-positive were selected.

### 2.2. Strain Handling

Each of the isolated Gram-positive *Bacillus* strains was cultivated in MRS broth at 37 ± 2 °C for 24 h, with a pass of 2% (*v*/*v*) in the same medium incubated for 12 h. The cells were collected by centrifugation at 5000× *g* for 10 min at 4 °C (Centurion Scientific K2015 centrifuge, UK), resuspended in MRS broth and skim milk at 10% (*w*/*v*), ultra-frozen at −50 °C (Thermo Scientific Revco Ultima II, USA), and freeze-dried (Labconco^®^ FreeZone 4.5L, KC, USA) for their preservation at the long term and in MRS-glycerol broth (50–50% *v*/*v*) for their use in subsequent tests.

*Lacticaseibacillus rhamnosus* ATCC 53103, *Lactiplantibacillus plantarum* ATCC 8014, and *Lactococcus lactis* ATCC 0205V reactivated in MRS broth for 24 h and a 2% (*v*/*v*) pass incubated for 12 h at 37 ± 2 °C, were used as reference strains.

*Listeria monocytogenes* ATCC 15313 was cultivated twice in BHI broth (Brain Heart Infusion, Difco^TM^), incubated at 35 ± 2 °C for 24 h with a 2% (*v*/*v*) pass for 12 h. *Pseudomonas aeruginosa* ATCC 9027 was cultivated in Pseudomonas P agar (Difco^TM^) at 30 ± 2 °C for 24 h. *Escherichia coli* REPEC O103 isolated from rabbit kidneys [2], donated by Dr. Fernando Navarro-García (CINVESTAV-IPN, México), from a pure culture in LB agar (Luria Bertani, Difco^TM^), was incubated in LB broth at 37 ± 2 °C for 24 h and with a 2% (*v*/*v*) pass for 12 h in an orbital shaking incubator (Lab-Line^®^ 4628, USA). *Campylobacter jejuni* subsp. *jejuni* ATCC 49943 was propagated in Soy Trypticase agar (Difco^TM^) with 5% (*v*/*v*) bovine defibrinated blood, incubated at 37 ± 2 °C for 48 h with an Anaerocult^®^ C system (Millipore^®^, USA) 6–16% O_2_, 7–10% CO_2_, followed by a Brucella broth (Difco^TM^) pass, incubated under the same conditions.

### 2.3. Microbiological Tests

The isolated strains were identified by colonial morphology in an agar plate, Gram stain, and cellular morphology by optical microscopy (Olympus^®^ CX31, USA). Catalase and oxidase tests were performed using *L. rhamnosus* ATCC 53103, *L. plantarum* ATCC 8014, and *Lc. lactis* ATCC 0205V as a negative control, and *P. aeruginosa* ATCC 9027 (catalase) and *S. aureus* ATCC 6538P (oxidase), respectively, as positive controls. The oxidative and fermentative metabolism was determined with the carbohydrate oxidation-fermentation test [23]: 1% (*w*/*v*) glucose, 1% (*w*/*v*) sorbitol, 1% (*w*/*v*) xylose, 2% (*w*/*v*) lactose and 0.5% (*w*/*v*) saccharose.

### 2.4. Molecular Identification

#### 2.4.1. Genomic DNA Extraction

Twelve-hour cultures from each one of the LAB strains were carried out for DNA extraction through a phenol-chloroform method [24], verifying its purity and concentration in a spectrophotometer (NanoDrop^®^ ND-1000, USA) (A_260–280 nm_), in addition to its integrity in agarose gels at 1.5%. The DNA of *L. lactis* ATCC 0205V and *L. rhamnosus* ATCC 53103 strains equally cultivated were used as control.

#### 2.4.2. Primer Sequences Design

Species-specific primers for BAL strains were designed (Table 1) from the strains reported in the literature, usually found in mouse intestinal microbiota. The partial or total sequences of the 16S rRNA gene were obtained from the GenBank from each of these strains. These sequences were used for the design of primers in the Primer-BLAST software by setting the following parameters: 18 to 25 nucleotides length, percentage of guanine-cytokine (%GC) between 40% and 50%, and difference of fusion temperature less than or equal to 5 °C. The designed primers were synthesized by Sigma Aldrich Química S de RL de CV.

#### 2.4.3. PCR Conditions

A PCR reaction was conducted with a final volume of 25 μL according to Master Mix (Promega^©^) kit specifications and 5 μL of the DNA sample (~150 ng) under the following conditions: initial denaturation at 95 °C for 5 min, followed by 35 cycles at 95 °C for 30 s, with annealing temperature for each primer (Table 1) for 30 s, 72 °C for 30 s and one final extension at 72 °C for 5 min. PCR products were resolved through electrophoresis in agarose gels at 2% (*w*/*v*), at 90 V for 1 h, using GeneRuler^TM^ (DNA Ladder Mix) as molecular weight standard, stained with ethidium bromide and visualized under UV light. Enzymatic cleaning was performed with Exo-Sap IT^TM^ (Applied Biosystems, USA) for sequencing at the Laboratory of Molecular Biochemistry in UBIPRO, FES Iztacala, UNAM, using the Sanger method.

#### 2.4.4. Sequence Analysis and Identity Percentage Matrix

The 16S rRNA sequences of each bacterial isolation obtained from the chromatograms were edited with BioEdit 7.2 software, and the sense and antisense sequences of each strain were aligned with ClustalW to create their consensus sequence. Every consensus sequence was aligned with BLAST from NCBI and compared with the sequences from the GenBank database. The similarity percentage between the sequences was obtained with multiple alignments in ClustalW and the option of an identity matrix.

#### 2.4.5. GenBank Submission

The *Ligilactobacillus murinus* sequences were submitted in GenBank and obtained the accession numbers: LGM A1 (OK668196); LGM A2 (OK668197); LGM B1 (OK668198); LGM C1 (OK668199); LGM D1 (OK668200); LGM E1 (OK668201); LGM E2 (OK668202); LGM E3 (OK668203); LGM E5 (OK668204); LGM F1 (OK668205); LGM F2 (OK668206); LGM F3 (OK668207); LGM G1 (OK668208); LGM H1 (OK668209); LGM H3 (OK668210) and LGM I1 (OK668211).

#### 2.4.6. Phylogenetic Tree Building

The sequences of the 18 isolated LAB strains were used for the building of phylogenetic trees, in addition to 12 reference sequences obtained from the alignment in BLAST-NCBI software. The phylogenetic analysis was performed in MEGA11 software, with the Maximum-Likelihood method of the suggested Jukes-Cantor model [25]. Confidence levels were determined by bootstrap analysis with 1000 repetitions.

### 2.5. Antagonistic Activity by Agar Diffusion Tests

Inhibitory activity was detected by diffusion tests in agar plates [26], with some modifications. MRS-based agar plates (BD, Difco^TM^) at 1.5% (*w*/*v*) were used, upon which 5μL aliquots of a logarithmic phase culture of each isolated LAB were placed, corresponding to 7–9 log CFU mL^−1^. In parallel, for each test, a 10 mL soft agar overlay (0.6% *w*/*v*), BHI agar for *L. monocytogenes*, LB agar for *E. coli*, and Brucella agar for *C. jejuni* was prepared, added with 100 μL from a 12 h culture of each test pathogen. Each overlay was conducted at 45 °C, and it was poured over every MRS-based agar plate. Solidified plates were incubated at 37 ± 2 °C for 24 h, and the test was conducted with *C. jejuni*. The plates were incubated with the use of the Anaerocult^®^ C system (5–7% O_2_, 8–10% CO_2_) at 40 ± 2 °C for 48 h.

### 2.6. Bacterial Growth Parameters

The growth kinetics of every isolated strain was created from new cultivation in MRS broth at 37 ± 2 °C, and OD and CFU mL^−1^ were obtained at 0, 4, 8, 12, 16, 20, and 24 h. Growth parameters were calculated with the CFU mL^−1^ from the logarithmic phase and the following formulas [27]:

Specific growth rate (μ): means the growth rate per unit of CFU or biomass
μ = 2.3·(Log X − Log Xo)/t − to

Generation time (τ): the time required to form one generation of cells (time between two divisions).
τ = Log 2·(t − to/Log X − Log Xo)

### 2.7. Supernatant Bacteriolytic Activity

#### 2.7.1. Obtention of Supernatants Culture

Cultures of every LAB strain in logarithmic phase (4 h), in 1 mL MRS broth, in Eppendorf tubes, were centrifuged at 5000× *g*, for 10 min at 4 °C (Centurion Scientific K2015, UK), neutralized at pH 7 (NaOH 6N) and sterilized by membrane filtration 0.22 µm (Merck Millipore^TM^, Ireland) [28].

#### 2.7.2. Protein Quantification

Proteins from Sterile-filtered, neutralized supernatants were precipitated with the Methanol/Chloroform method [29]. The protein concentration of each sample was determined by the Bradford method [30], with Bradford reactive (Bio-Rad, cat. no. 5000205, USA). In every test, a standard curve was built via Bovine Seric Albumin 1 mg·mL^−1^ (Bio-Rad cat no. 5000206).

#### 2.7.3. Electrophoretic Profile of Supernatants Proteins (SDS-PAGE)

Electrophoresis was performed on denaturing and reducing polyacrylamide gels at 12% (30%T,2.67%C) [31]. Fifteen µL of the sample containing the precipitated proteins (1 µg/µL) and 5 µL of Laemmli sample buffer 4× (Bio-Rad, cat no. 1610747) were heated to 95 °C for 5 min and placed in each well of the gel. Precision Plus Protein^TM^ All Blue molecular Weight marker (Bio-Rad, cat. no. 1610373) was used. The gels were run in an electrophoresis chamber (MiniPROTEAN III, Bio-Rad) at 90 volts for 180 min and 20 mA (Bio-Rad, PowerPac Basic). After electrophoresis, the gels were washed with sterile distilled water, stained with Coomassie brilliant blue (Bio-Rad cat. no. 1610400) for 1 h, and destained with a solution (20% methanol (J.T. Baker^®^, USA), 15% acetic acid (J.T. Baker^®^, USA) and 65% deionized water) for 12 h. Gels were analyzed in the Gel Logic 100 imaging system (Kodak, USA) to establish the electrophoretic profile of protein bands.

#### 2.7.4. Zymograms

The zymography tests were carried out with the cells obtained from 60 mL of a culture of *L. monocytogenes*, *E. coli* REPEC, or *C. jejuni*, washed twice with sterile saline solution (0.9% *w*/*v*), which were used as a substrate when adding them in 12% polyacrylamide mixtures. *Micrococcus lysodeikticus* ATCC 4698 was used as a positive control at 0.2% (*w*/*v*) for the identification of PGH activity [32]. Precipitated proteins of every LAB strain were placed in the wells, and the SDS-PAGE was carried out under the same conditions previously described. When the electrophoresis was finished, the gels were disassembled and put in a container with deionized water for 30 min, proceeded by incubation in a renaturing solution (25 mM Tris-HCl pH 8 with 1% *v*/*v* Triton) for 6 h with the purpose of identifying lytic bands. In order to obtain a better definition, the gels were stained with methylene blue at 1% (*w*/*v*) in KOH solution at 0.1% (*w*/*v*) for 30 min and were subjected to destaining with distilled water [33].

#### 2.7.5. UPLC-MS Fingerprinting of *L. murinus* B1 Lytic Proteins

The protein bands that demonstrated lytic activity separated in SDS-PAGE gels, stained with Coomassie brilliant blue G-250 0.02% (*w*/*v*), were cut out of the gel with a sterile scalpel and transferred to an Eppendorf tube. The preparation of the samples was carried out according to the protocol developed at the Research and Industry Support Services Unit (USAII), Faculty of Chemistry, UNAM [34]. The gel fragments were incubated in 100 μL of destaining solution (80 mg ammonium bicarbonate, 20 mL analytically pure water (J.T. Baker^®^), and 20 mL acetonitrile) for 15 min in an orbital shaker at 50 rpm.

To reduce and alkylate the proteins in the samples, the liquid phase was discarded, and 5.5 μL of 0.5 M TCEP (Tris-(2-Carboxyethyl)phosphine) and 50 μL of 25 mM ammonium bicarbonate were added for 10 min at 60 °C. The liquid phase was discarded and alkylated with 200 μL of 200 mM iodoacetamide (J.T. Baker^®^) for 1 h in a dark tube. The samples were washed with 200 μL of destaining solution for 5 min, then dehydrated with 200 μL of 100% acetonitrile (J.T. Baker^®^) for 15 min at 24 °C and dried for 5 min in a vacuum concentrator (Genevac^TM^).

Enzymatic digestion was performed with 1 μL of trypsin (Promega cat #V528A) (1μg/μL) and 100 μL of 25 mM ammonium bicarbonate, incubated for 18 h at 37 °C. Peptides were extracted with 50% (*v*/*v*) acetonitrile and 5% formic acid (J.T. Baker^®^), and the sample placed in a new tube was dried in a vacuum concentrator for 2 h. Peptides were resuspended in 10 μL of 0.1% (*v*/*v*) formic acid, and sample salts were removed with a protein concentrator (Bond Elut OMIX, C18, No. A5700310K) and eluted in 25 μL of mobile phase (3% acetonitrile, 97% analytically pure water and 0.1% formic acid).

Peptide analysis was performed on an SYNAPT G2-S integrated nano-LC-ESI-/MS system (Waters Corporation, UK) equipped with a NanoLockSpray ion source and coupled online to a nanoACQUITY Ultra Performance Liquid Chromatograph (UPLC; Waters Corporation). The binary solvent system was composed of 2% acetonitrile with 0.1% formic acid (mobile phase A) and 98% acetonitrile with 0.1% formic acid (mobile phase B). The sample was injected onto a nanoACQUITY UPLC 2G Trap column, 5 µm, 180 µm × 20 mm (S/N 03073720616583) and subsequently washed with mobile phase A at a flow rate of 0.4 µL/min. Peptides were separated on a nanoACQUITY UPLC HSS T3 C18 column, 1.8 µm, 75 µm × 150 mm (S/N 01973716416209) using a linear gradient to 40% mobile phase B with a flow rate of 0.4 µL/min.

Data were acquired and processed using Protein Lynxs Global Selver 2.5.1^TM^ software, Waters (PLGS). For the identification of the proteins, the databases extracted from the website were used: https://www.uniprot.org, accessed on 5 January 2023, for *Ligilactobacillus*.

### 2.8. Statistical Analysis

Three independent replicates of each assay were performed in duplicate for each one. A variance analysis ANOVA was applied to the obtained data, expressed as means and standard deviation. The Tukey test was conducted for the comparison of the media with a significance level (*p* < 0.05) with the Minitab^®^ statistical software.

## 3. Results

### 3.1. LAB Isolation and Microbiological Tests

Eighteen LAB strains were isolated under anaerobiosis conditions from the intestine of just-weaned CD1 mice per intestine section with a distribution of 28% duodenum (5 isolations), 22% jejunum (4 isolations), and 50% colon (9 isolations). In biochemical tests, the 18 isolated strains were negative for catalase and oxidase. In oxidation-fermentation tests, all strains fermented glucose, while only E4 and H2 strains fermented sorbitol. Moreover, A2, C1, and D1 strains were negative to xylose; E1, F3, and H1 strains were negative to lactose, and D1, E1, F2, and H1 strains were negative to saccharose.

### 3.2. Molecular Identification

Amplicons of 224 pb were obtained with the primers for *L. animalis* (Appendix A). The results of the alignment in BLAST-NCBI identified 16 strains as *Ligilactobacillus murinus*, of which the D1, E2, and G1 strains displayed 100% identity, while 13 others shared similarity percentages between 97.89% and 99.51%. The H2 strain was identified as *Ligilactobacillus animalis* (95.10%), and the E4 strain as *Streptococcus salivarius* (97.54%) (Appendix A).

#### 3.2.1. Identity Percentage Matrix

The identity matrix allowed the authors to establish the similarity of the isolated BAL sequences. As a result, 100% similarity was obtained between *L. murinus* C1 and E1 and between *L. murinus* E2 and D1. The similarity for the rest of the *L. murinus* strains was between 92% and 99% (Table 2).

#### 3.2.2. Phylogenetic Tree

The phylogenetic tree built by the Maximum-Likelihood based on the Jukes-Cantor model shows that strains with similar sequences are clustered in the same clade level [35]. In particular, the isolated *L. murinus* E3 and I1 strains in this study were clustered in the same clade as the *L. murinus* DSM 100193, TCD6, NM28, and V10 (isolated from mice), *L. animalis* P38 and FR12 (isolated from chicken) and *L. animalis* JCM8692 (isolated from pork) strains found in GenBank. Likewise, the remaining *L. murinus* and *L. animalis* H2 strains isolated in this study were located in different clades (Figure 1).

### 3.3. Bacterial Growth Parameters

The growth parameters for every strain are listed in Table 3. *L. murinus* E3 and *S. salivarius* E4 displayed the highest μ values with 1.25 h^−1^ and 1.13 h^−1,^ respectively; on the other hand, *L. murinus* D1, E5, H1, and H3 were clustered with intervals between 0.39 and 0.51 h^−1^. Regarding τ, *L. murinus* H3 had the highest value at 2.12 h, and the lowest values were observed for *L. murinus* A1, A2, E3, and *S. salivarius* E4 with intervals between 0.59 and 0.84 h.

### 3.4. Agar Diffusion Tests

The inhibitory activity against *L. monocytogenes* was observed in the 18 isolated strains, highlighting the activity of *L. animalis* H2 with inhibition diameters of 17 mm (*p* < 0.5) and *L. murinus* A1, A2, B1, C1, E2, F1 and F3 with inhibition diameters of 15 mm. Additionally, the isolated LAB strains showed an inhibitory effect against *E. coli* REPEC, with larger inhibition halos for *L. animalis* H2 with 25 mm and *L. murinus* G1, A1, and F3 with 23 mm. During the tests performed with isolated LAB strains against *C. jejuni*, a significant inhibitory effect was observed, highlighting the activity of *L. animalis* H2 with inhibition diameters of 44 mm (*p* < 0.5) (Table 4).

### 3.5. SDS-PAGE and Zymograms

As a result, in protein profiles, some differences were observed in the number and intensity of protein bands. Zymograms against the different testing microorganisms evidenced two bands with an apparent molecular mass of 108 and 25 kDa, corresponding to two lytic bands of 107 and 24 kDa. The 107 kDa lytic bands were identified in the 16 *L. murinus* strains against *M. lysodeikticus* ATCC4698, *E. coli* REPEC O103, and *C. jejuni.* Moreover, the lytic band was identified in renatured zymograms after two hours against *C. jejuni* ATCC 49943, and after renaturation, for six hours, the lytic activity was observed as diffuse. On the other hand, the 24 kDa lytic bands were detected with different intensities in the *L. murinus* strains against *L. monocytogenes*, *E. coli* REPEC O103, and *C. jejuni*. It is important to highlight that the 107 kDa lytic band found in *L. murinus* B1 supernatants displayed a higher density against *E. coli* REPEC, and 24 kDa lytic bands in *L. murinus* H3 and I1 supernatants against *L. monocytogenes* (Figure 2). *L. animalis* and *S. salivarius* strains did not show lytic activity against the testing microorganisms.

### 3.6. UPLC-MS Analysis

The UPLC-MS analysis confirmed the presence of several specific peptides obtained from the protein bands with lytic activity (Figure 3). Through the detection of 20 peptides (Table 5), the lytic band with a molecular mass of 107 kDa was identified as a PGH with N-acetylmuramoyl-L-alanine amidase activity. This PGH has a molecular structure made of a catalytic domain formed by two 1,4-beta-N-acetylmuramoylhydrolase regions and one domain bound to the peptidoglycan formed by nine repeated sequences of amino acids called LysM (lysin-like motif) domains (UniProt accession A0A4S2EQ32_9LACO).

Eight peptides were identified in the 24 kDa lytic band via the UPLC-MS analysis. Such peptides showed similarities with 12% of coverage with a protein fragment containing the GA (protein G-related Albumin-binding) domain within the genomic sequence of *L. salivarius* with aminopeptidase molecular function (UniProt accession A0A1V9RCV3_9LACO).

## 4. Discussion

Several niches were explored to obtain LAB, including human and animal gastrointestinal tract [16,17]. In this study, LAB strains were isolated under anaerobiosis conditions from recently weaned CD1 mice intestines. The 18 catalase and oxidase-negative isolated strains showed differences in their fermentation profile. All the strains fermented the glucose (hexose) used by LAB as the main source of energy. Meanwhile, most strains fermented pentose (xylose) and disaccharides (lactose and saccharose). On the other hand, the strains identified as *L. murinus* did not ferment sorbitol (polyol sugar). These results match those reported for the *L. murinus* 313 reference strain in terms of glucose, lactose, and saccharose fermentation, which does not ferment xylose and sorbitol [36]. Based on the fermentation type, lactobacilli are classified into three groups: obligately homofermentative, fermenting via the glycolytic pathway; facultative heterofermentative, fermenting via the glycolytic/pentose phosphoketolase pathway and obligately heterofermentative, fermenting via the pentose phosphoketolase pathway. Therefore, obligately homofermentative lactobacilli cannot assimilate pentoses [37]. However, there are exceptions for some strains, such as the homolactic fermentation of a pentose, for instance, phosphoketolase is induced in the presence of xylose, and the metabolism may shift between these two pathways according to the xylose concentration in the medium [38].

Molecular identification allowed the detection of a high proportion of *L. murinus* and one *L. animalis*. Specifically, *L. murinus* strains have been isolated from different niches, such as rodent gastrointestinal tract [39], dog feces [40], fermented dough [41], and marine sediments [19], while *L. animalis* has been found in the gastrointestinal tract of several animal species [42]. Furthermore, the identity matrix and the phylogenetic analysis determined the similarity between the sequences using the V2 variable region of the 16S rRNA gene from the 16 *L. murinus* strains and confirmed its close relationship with *L. animalis*, which has been described by several authors, identifying these two species at the same phylogenetic level with similarity percentages of up to 99% between 16S rRNA gene regions [41,43].

The isolated LAB inhibitory activity against the testing pathogens was demonstrated by the agar diffusion method. *L. animalis* H2 and *L. murinus* A1, A2, B1, C1, E2, F1 and F3 strains displayed more inhibition (*p* < 0.5) against *L. monocytogenes*, while *L. animalis* H2 and *L. murinus* G1, A1 and F3 showed larger halos against *E. coli* REPEC. In these studies, the inhibitory activity of the 18 isolated strains was highlighted against *C. jejuni*. According to reports, several LABs have displayed antagonistic activity against *Campylobacter* strains. For instance, *Pediococcus pentosaceus* CWBI B73, *Lactiplantibacillus pentosus* CWBI B78, and *Enterococcus faecium* THT inhibited *C. jejuni* and *C. coli* species [44]. On the other hand, *L. salivarius* SMXD51, MMS122, and MMS151 showed inhibition capacity against *C. jejuni* and *C. coli* [45]. In these studies, besides evaluating the antagonistic activity of LAB through the agar-diffusion method, cell supernatants were assessed, confirming that the inhibition capacity of the isolated strains was related to the production of protein compounds.

In general, antibacterial compounds with a protein nature are produced by LAB in low quantities, which makes them difficult to detect and analyze. Therefore, in this study, SDS-PAGE electrophoretic and zymography techniques were used, which are highly sensitive methods for the detection of enzymatic activity [46], which showed the presence of two proteins with lytic activity existing in the supernatants of the 16 strains of *L. murinus* isolated.

The 107 kDa lytic band, found in copolymerized gels with *M. lysodeikticus*, *E. coli* REPEC O103, and *C. jejuni*, was identified via the UPLC-MS fingerprinting method as an extracellular PGH with N-acetylmuramoyl-L-alanine amidase activity, considered as a bacteriolytic enzyme with antibacterial activity. According to the literature, the PGH are enzymes that hydrolyze peptidoglycan (PG) links, known as the main component of the cellular wall for Gram-positive and Gram-negative bacteria. The PG is formed by chains of alternate N-acetylglucosamine (GlcNAc) and N-acetylmuramic acid (MurNAc) units, bound by B-1,4 links and cross-linked by peptides formed out of L- and D-amino acids. Specifically, the bacteriolytic activity from N-acetylmuramoyl-L-alanine amidase originated due to the specific site of hydrolysis in the PG, which cleaves the amide link between MurNAc and the L-alanine of the peptide [47]. Furthermore, the efficiency of the N-acetylmuramoyl-L-alanine amidase activity identified can be attributed to the number of binding LysM modules, which recognize the GlcNAc scraps of the PG and contribute to their additive junction [48].

The 24 kDa band with lytic activity against *L. monocytogenes*, *E. coli* REPEC O103, and *C. jejuni* displayed 12% similarity with a GA domain-containing protein fragment, reported in the *L. salivarius* genome. Particularly, the GA domain identified in this lytic band via the IDQMLELTVDQKDNFNK peptide is a domain localized in a variety of proteins of the cellular surface in different bacteria [49].

Similar to this research, several studies have reported the lytic activity from PGH against Gram-positive and Gram-negative pathogens [28,50]. One of the most researched PGH is lysostaphin, a 25 kDa peptidase produced *by Staphylococcus simulans*, that disrupts the peptide link between the third and fourth glycine residues of the pentaglycine cross-link in the *S. aureus* PG, which was demonstrated to be efficient in infections caused by *Staphylococcus* spp. [51].

The findings of this study can be applied from the perspective of the sustainable health and well-being objective [7] and the multidisciplinary effort with the One Health framework to reduce health risks such as sanitary problems, zoonotic illnesses, and antimicrobial resistance because these issues represent a growing threat for human and animal well-being. In the search for alternative solutions, there is the use of new probiotic strains and their bioactive components to reduce the use of antimicrobials because they provide health benefits and do not generate risks of resistance mechanisms. In this context, our research is important because it explores a new niche in the pursuit of *Lactobacillus* strains, which displayed inhibitory capacity against test pathogen microorganisms, *C. jejuni*, which is considered the main bacterial cause of gastroenteritis in humans around the world and the zoonotic pathogen responsible for gastroenteritis in humans associated to chicken meat consumption [52]. *E. coli* EPEC was identified as one of the main causes of diarrhea in humans and animals [2] and acknowledged as the primary bacteria that causes diarrhea in children under five years old [53], and *L. monocytogenes* considered as one emergent Food-borne pathogen, provokes severe health issues that appear sporadically or as an epidemic [54]. Due to the inhibitory capacity of the 16 *Ligilactobacillus murinus* isolated strains presented in this study, as well as the identification of their antibacterial metabolites, a feasible alternative is suggested for the prevention and control of these pathogens in order to promote animal health and well-being and, subsequently, human health as well.

## 5. Conclusions

The molecular identification of 16 strains of *Ligilactobacillus murinus* isolated from mice intestinal tract, its wide spectrum antibacterial activity against clinical and zoonotic pathogens, and the presence of bacteriolytic enzymes, one of them being PGH are important finding that contributes to the antagonist potential of these strains. In addition, these enzymes have a potential application as alternative antibacterial agents and treatment of infections caused by antibiotic resistant bacteria because the molecular mechanisms of resistance to antibiotics are not related to the hydrolysis of the PG.

## Figures and Tables

**Figure 1 microorganisms-11-00942-f001:**
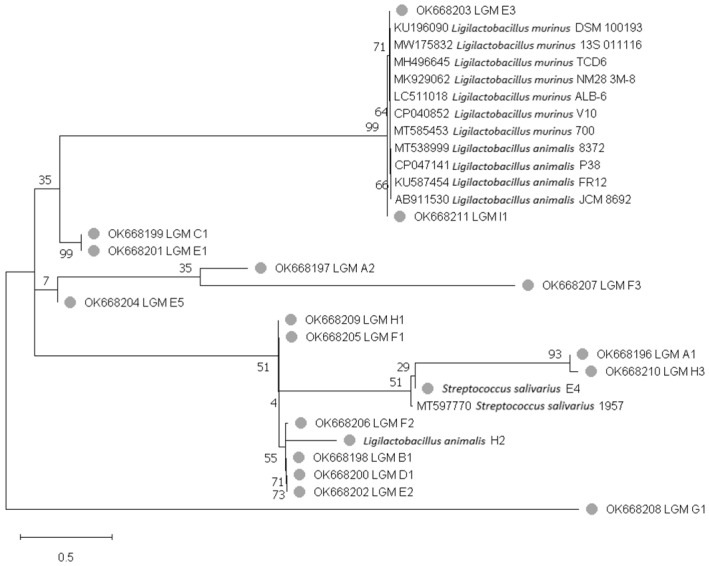
Phylogenetic tree built by the Maximum-Likelihood method, using the sequences of V2 variable region of the 18 LAB strains isolated from mice intestinal tract (●), compared with strains found in GenBank. The numbers expressed before each clade (shown in percentages) show the trust levels generated from 1000 bootstrap.

**Figure 2 microorganisms-11-00942-f002:**
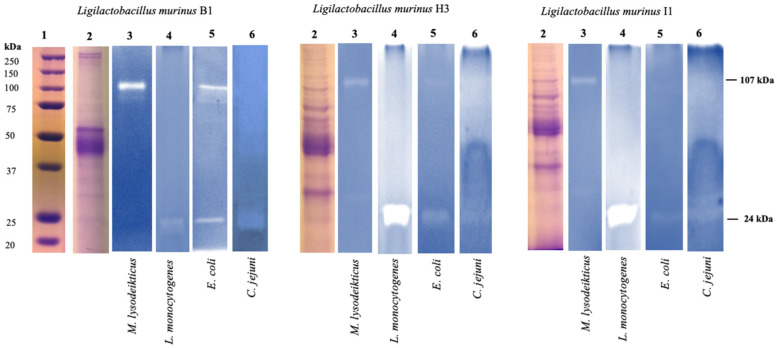
SDS-PAGE, 12%, and Zymograms. Line 1 Molecular Weight Marker, Precision Plus Protein^TM^ All Blue, Bio-Rad. Line 2 Protein profile of culture supernatant in logarithmic phase. Zymograms copolymerized with Line 3 *Micrococcus lysodeikticus*, Line 4 *Listeria monocytogenes*, Line 5 *Escherichia coli* REPEC, and Line 6 *Campylobacter jejuni*.

**Figure 3 microorganisms-11-00942-f003:**
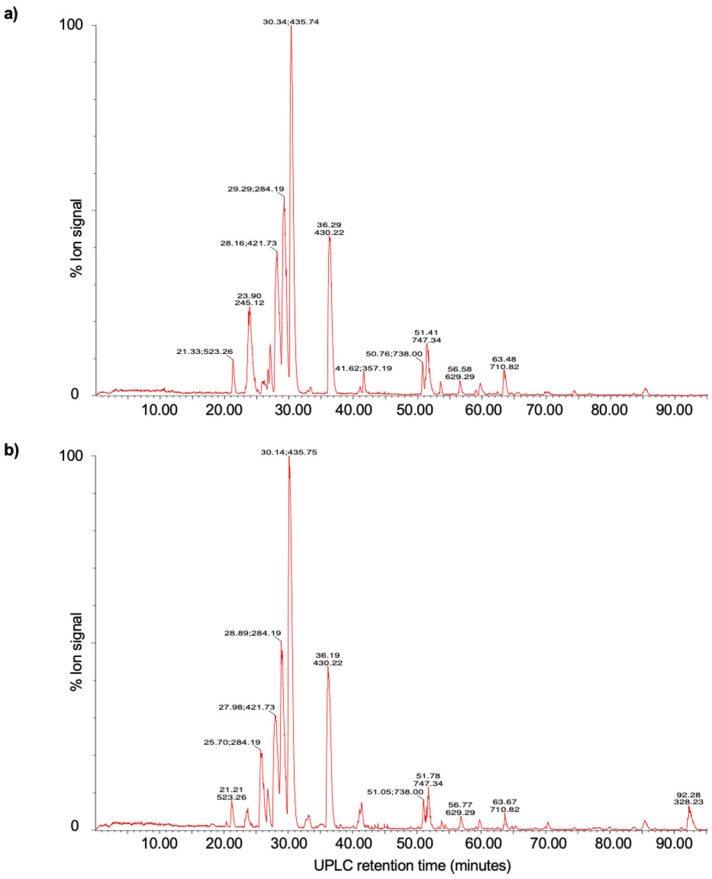
UPLC-MS of the SDS-PAGE gel bands with lytic activity, (**a**) protein with an apparent molecular mass of 107 kDa, and (**b**) 24 kDa. The masses in the spectra are protonated masses of the most prominent detected compounds.

**Table 1 microorganisms-11-00942-t001:** Species-specific primers were used for the identification of isolated LAB strains from the mouse intestinal tract.

LAB Name	PrimerID	Sequence (5′→3′)	Amplicon Size (bp)	Annealing Temp. (°C)
*Lactobacillus intestinalis*	Lbinte-F	GTACAACGAGAAGCGAGCCT		
	Lbinte-R	CACATAAGTGGTTAGGCCACC	213	58
*Ligilactobacillus animalis*	Lbanim-F	GAGTAACACGTGGGCAACCT		
	Lbanim-R	TGTCTCAGTCCCAATGTGGC	224	57
*Limosilactibacillus reuteri*	Lbreute-F	AGTCACGGCTAACTACGTGC		
	Lbreute-R	TTCGGTTAAGCCGAGTTTCCA	127	57
*Lactobacillus delbrueckii* spp. *bulgaricus*	Lbdelb1-F	CCAAGGCAATGATGCGTAGC		
	Lbdelb1-R	TTGCTCCATCAGACTTGCGT	127	56
*Lactobacillus acidophilus*	Lbacid-F	ACGTCAAGTCATCATGCCCC		
	Lbacid-R	TTAGACGGCTCCTTCCCGAGT	277	59
*Limosilactobacillus fermentum*	Lbferm-F	TCTTGCGCCAACCCTAGAGA		
	Lbferm-R	GACTCGGTGTTTGGGTGTTACAAAC	446	63
*Lactococcus lactis* spp. *lactis*	Lclact1-F	GAGCGCTGAAGGTTGGTACT		
	Lclact1-R	TGTCTCAGTCCCAATGTGGC	272	57
*Lactococcus lactis* spp. *cremoris*	Lclact2-F	GGCGTGCCTAATACATGCAA		
	Lclact2-R	CCGTTCGCTGCTCTTCAAAT	70	55
*Lactobacillus gasseri*	Lbgass-F	GAGCGAGCTTGCCTAGATGA		
	Lbgass-R	CTCTAGACATGCGTCTAGTGTT	163	58
*Ligilactobacillus murinus*	Lbmuri-F	AAGAGTTGAGCTGAGCGAACG		
	Lbmuri-R	CGTAGAAGTTTGGGCCGTGTTT	268	60
*Lactobacillus crispatus*	Lbcris-F	GTACCAAGCCAAAGCAAGAC		
	Lbcris-R	GTTTGAAGCCTTTACGTAAGTC	383	55
*Lactiplantibacillus plantarum*	Lbplan-F	CCGTTTATGCGGAACACCTA		
	Lbplan-R	TCGGGATTACCAAACATCAC	318	55

**Table 2 microorganisms-11-00942-t002:** Identity percentage matrix. Similarity values of the V2 variable region of the 16S rRNA gene of the isolated LAB.

	I1	H3	H2	H1	G1	F3	F2	F1	E5	E4	E3	E2	E1	D1	C1	B1	A2	A1
*L. murinus* A1	96.0	99.0	89.8	97.5	94.2	94.0	98.0	97.0	93.2	71.4	97.0	96.5	97.5	96.5	97.5	97.0	97.5	100
*L. murinus* A2	94.6	97.0	87.9	96.0	92.7	93.5	96.0	96.5	93.1	71.0	95.1	96.0	96.0	96.0	96.0	96.5	100	
*L. murinus* B1	96.5	97.5	89.8	98.0	94.6	92.1	97.5	98.5	92.2	72.4	97.0	99.5	98.0	99.5	98.0	100		
*L. murinus* C1	98.0	97.5	90.7	98.0	96.0	92.1	97.5	98.5	92.7	72.0	99.0	97.5	100	97.5	100			
*L. murinus* D1	96.0	97.0	89.8	98.5	94.6	91.6	97.0	99.0	92.2	72.4	96.5	100	97.5	100				
*L. murinus* E1	98.0	97.5	90.7	98.0	96.0	92.1	97.5	98.5	92.7	72.0	99.0	97.5	100					
*L. murinus* E2	96.0	97.0	89.8	98.5	94.6	91.6	97.0	99.0	92.2	72.4	96.5	100						
*L. murinus* E3	99.0	97.0	91.2	97.5	97.0	91.7	97.0	97.5	92.2	71.2	100							
*S. salivarius* E4	71.1	71.9	67.4	71.9	70.3	69.6	71.9	72.4	68.9	100								
*L. murinus* E5	92.2	94.1	86.6	93.6	91.3	90.2	93.2	93.1	100									
*L. murinus* F1	97.0	97.5	90.2	99.5	95.6	92.1	97.5	100										
*L. murinus* F2	98.0	99.0	90.7	98.0	95.1	93.6	100											
*L. murinus* F3	91.7	94.5	87.8	92.6	89.9	100												
*L. murinus* G1	97.0	95.1	91.2	95.6	100													
*L. murinus* H1	97.0	98.0	90.7	100														
*L. animalis* H2	91.2	90.7	100															
*L. murinus* H3	97.0	100																
*L. murinus* I1	100																	

**Table 3 microorganisms-11-00942-t003:** Kinetic parameters of the isolated LAB strains. Data are expressed as a mean and standard deviation. ^abcdef^ shows a significant difference per column (Tukey).

Isolated LAB	Specific Growth Rate (μ) h^−1^	Generation Time (τ) h
1. *L. murinus* A1	0.82 ± 0.04 ^cd^	0.84 ± 0.04 ^cd^
2. *L. murinus* A2	0.91 ± 0.11 ^bc^	0.77 ± 0.06 ^cd^
3. *L. murinus* B1	0.60 ± 0.02 ^def^	1.11 ± 0.15 ^bcd^
4. *L. murinus* C1	0.74 ± 0.03 ^cde^	0.93 ± 0.04 ^bcd^
5. *L. murinus* D1	0.51 ± 0.06 ^ef^	1.43 ± 0.32 ^b^
6. *L. murinus* E1	0.56 ± 0.09 ^def^	1.19 ± 0.18 ^bc^
7. *L. murinus* E2	0.78 ± 0.04 ^cde^	0.88 ± 0.05 ^bcd^
8. *L. murinus* E3	1.25 ± 0.27 ^a^	0.59 ± 0.14 ^d^
9. *S. salivarius* E4	1.13 ± 0.18 ^ab^	0.62 ± 0.13 ^d^
10. *L. murinus* E5	0.50 ± 0.06 ^ef^	1.26 ± 0.23 ^bc^
11. *L. murinus* F1	0.60 ± 0.04 ^def^	1.15 ± 0.09 ^bcd^
12. *L. murinus* F2	0.79 ± 0.01 ^cde^	0.91 ± 0.06 ^bcd^
13. *L. murinus* F3	0.54 ± 0.06 ^def^	1.41 ± 0.14 ^b^
14. *L. murinus* G1	0.72 ± 0.02 ^cde^	0.93 ± 0.06 ^bcd^
15. *L. murinus* H1	0.50 ± 0.06 ^ef^	1.43 ± 0.26 ^b^
16. *L. animalis* H2	0.54 ± 0.04 ^def^	1.28 ± 0.18 ^bc^
17. *L. murinus* H3	0.39 ± 0.05 ^f^	2.12 ± 0.36 ^a^
18. *L. murinus* I1	0.56 ± 0.06 ^def^	1.29 ± 0.28 ^bc^

**Table 4 microorganisms-11-00942-t004:** Inhibition halos on isolated LAB strain agar plates (mm). Data are expressed as means and standard deviation. ^abc^ for columns indicates a significant difference (Tukey).

	Inhibition Diameter (mm)		
Isolated LAB	*L. monocytogenes*	*E. coli*	*C. jejuni*
1. *L. murinus* A1	15 ± 0.5 ^ab^	21 ± 1.2 ^abc^	36 ± 3.7 ^b^
2. *L. murinus* A2	15 ± 0.1 ^ab^	19 ± 1.6 ^bc^	42 ± 1.0 ^ab^
3. *L. murinus* B1	15 ± 0.6 ^ab^	20 ± 1.0 ^bc^	39 ± 2.6 ^ab^
4. *L. murinus* C1	15 ± 0.6 ^ab^	20 ± 0.5 ^bc^	42 ± 2.5 ^ab^
5. *L. murinus* D1	14 ± 0.6 ^ab^	19 ± 1.5 ^bc^	41 ± 1.1 ^ab^
6. *L. murinus* E1	14 ± 0.6 ^ab^	19 ± 1.0 ^c^	38 ± 2.0 ^ab^
7. *L. murinus* E2	15 ± 1.1 ^ab^	19 ± 1.5 ^c^	41 ± 2.3 ^ab^
8. *L. murinus* E3	14 ± 1.1 ^ab^	18 ± 1.1 ^c^	41 ± 3.7 ^ab^
9. *S. salivarius* E4	14 ± 0.6 ^ab^	18 ± 1.1 ^c^	38 ± 1.5 ^ab^
10. *L. murinus* E5	14 ± 0.0 ^ab^	18 ± 1.5 ^c^	40 ± 2.6 ^ab^
11. *L. murinus* F1	15 ± 0.6 ^ab^	20 ± 1.8 ^bc^	41 ± 2.0 ^ab^
12. *L. murinus* F2	15 ± 0.6 ^ab^	20 ± 1.8 ^bc^	40 ± 2.0 ^ab^
13. *L. murinus* F3	15 ± 0.6 ^ab^	21 ± 0.5 ^abc^	41 ± 2.5 ^ab^
14. *L. murinus* G1	14 ± 1.0 ^ab^	23 ± 1.6 ^ab^	40 ± 1.5 ^ab^
15. *L. murinus* H1	14 ± 1.0 ^ab^	19 ± 1.1 ^bc^	41 ± 3.2 ^ab^
16. *L. animalis* H2	17 ± 0.6 ^a^	25 ± 1.5 ^a^	44 ± 1.7 ^a^
17. *L. murinus* H3	13 ± 0.1 ^b^	20 ± 1.2 ^bc^	36 ± 2.0 ^b^
18. *L. murinus* I1	14 ± 1.1 ^ab^	18 ± 1.1 ^c^	36 ± 2.0 ^b^

**Table 5 microorganisms-11-00942-t005:** Peptides obtained from the SDS-PAGE gel bands with lytic activity by UPLC-MS, (**a**) protein with an apparent molecular mass of 107 kDa and (**b**) 24 kDa.

Time	Sc	Sequence	Prec *m*/*z*	*z*	Prec MW	Theor MW	ΔMass
**(a) Peptidoglycan hydrolase OS *Ligilactobacillus murinus* OX 1622: PLGS Score 6997; coverage, 29%**							
69.95	7.49	TYPGNVQTFLNNIAGPAQQVAQQR	872.44	3	2615.32	2615.32	−1.87
64.18	7.90	VNNLSSDLIYVGQTLK	882.47	2	1763.95	1763.95	−1.33
42.18	7.78	AGDSLWAIANSHK	457.23	3	1369.68	1369.68	−0.36
72.07	7.58	SLNNLNSDLIFAGQVLK	923.50	2	1846.00	1846.00	−2.75
39.78	7.99	YGVYGTYATAPDYADK	877.89	2	1754.78	1754.78	−1.11
34.11	7.58	ANSANYAIAAQNLR	738.88	2	1476.75	1476.75	−0.36
62.06	7.61	NLNNLSSNLIMPGQVLK	928.00	2	1855.00	1855.00	−1.08
62.06	7.63	NLNNLSSNLIMPGQVLK	928.00	2	1855.00	1855.00	−1.08
28.11	7.65	AGDSLWR	402.70	2	804.39	804.39	−1.26
53.61	6.96	YSSYAESLNGYANVITTR	1004.9	2	2008.95	2008.95	−2.34
76.54	6.01	GAVTTANKPNTQSNTNK	873.46	2	1745.92	1745.92	24.50
56.39	8.52	NLNNLSSNLIMPGQVLK	936.00	2	1871.00	1871.00	−0.09
49.63	7.67	ALNNLTSDMIYVGQNLK	955.48	2	1909.96	1909.96	−1.43
69.20	0.00	VQTFLNNIAGPAQQVAQQR	689.36	3	2066.08	2066.08	3.36
28.11	0.00	DSLWR	676.34	1	676.34	676.34	−0.85
28.11	0.00	SLWR	561.31	1	561.31	561.31	0.35
42.43	0.00	AIANSHK	370.70	2	740.40	740.40	3.84
28.11	0.00	GDSLWR	733.36	1	733.36	733.36	−1.68
28.12	0.00	AGDSLWR	393.69	2	786.38	786.38	−1.73
42.19	0.00	AGDSLWAIANSHK	451.22	3	1351.67	1351.67	−2.99
**(b) GA domain-containing protein Fragment *Ligilactobacillus salivarius* OX 1624: PLGS Score 31.20; coverage, 12%**							
49.06	3.56	EPETPVNPSEPGK	690.83	2	1380.66	1380.66	2.87
53.41	3.40	EPETPVDPSEPEK	727.34	2	1453.67	1453.67	4.37
53.81	4.27	SPQELDQIFTGNNDTIDK	678.98	3	2034.94	2034.94	−6.90
48.88	3.85	EPETPVDPSESGKEPETPVDPSEPGK	684.32	4	2734.29	2734.29	14.27
49.06	3.56	EPETPVNPSEPGK	690.83	2	1380.66	1380.66	2.87
56.78	3.77	IDQMLELTVDQKDNFNK	684.34	3	2051.00	2051.00	−1.02
75.87	3.70	EIYLTGHSLGGYLAEYFAATK	768.74	3	2304.21	2304.21	25.52
51.72	3.92	ISVEFDPQYEYYKK	904.94	2	1808.87	1808.87	1.61

Sc, peptide score; prec m/z, precursor mass-to-charge ratio; z, charge; prec MW, precursor molecular Weight; theor MW, theoretical molecular weight. Data were processed using Protein Lynxs Global Selver 2.5.1^TM^ software, Waters (PLGS).

## Data Availability

The sequence data that support the findings of this study are openly available in GenBank (RRID:SCR_002760) at https://www.ncbi.nlm.nih.gov/genbank/, accessed on 3 November 2021, accession numbers: OK668196, OK668197, OK668198, OK668199, OK668200, OK668201, OK668202, OK668203, OK668204, OK668205, OK668206, OK668207, OK668208, OK668209, OK668210, OK668211. The data that support the findings of this study are available from the corresponding author upon reasonable request.

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
