# Peer review of "Ligilactobacillus murinus* Strains Isolated from Mice Intestinal Tract: Molecular Characterization and Antagonistic Activity against Food-Borne Pathogens"

_microorganisms, 2023, doi:10.3390/microorganisms11040942_

Round 1
Reviewer 1 Report
The authors present a coherent body of scientific information, collected using adequate methodology. However, they fail to demonstrate the importance of their work for the journal and their prospective readers. Why is it important/interesting to isolate potential probiotic bacteria from mice and apply them is these same animals? What potential further applications might these have? How could the findings of this study be further applied within the framework of the One Health concept? I don’t think the authors mean to give probiotics to mice as a means of preventing zoonoses, or to apply the strains they isolated in other animals/humans. Are they merely using mice as a model? How can their data from the mice model be useful to better understand the potential of probiotics as an aid in the control of food-borne diseases? Or their purpose is to highlight the potentialities of bacteriolytic proteins. This has to be clearly explained, otherwise the study will lack interest and attractivity to prospective readers. The authors need to rework their manuscript, in order to give a better insight into the interest and impact their research will have. The final part of the Introduction, as well as the Discussion and Conclusions, need to be rewritten in order to convey this information. The Abstract would also benefit from including, in a very concise way, information on the importance of this work.
Therefore, a considerable part of the manuscript needs to be rewritten, to emphasize the importance of the authors' findings, before it is ready for publishing in a Q1 journal such as Microorganisms.
Comments on specific parts of the manuscript are given below.
Line 41 – Food-borne should not have a capitalized initial.
Lines 45 – 48 – This sentence needs revising English. It is not grammatically correct, making it not clear at all. I guess what the authors mean to say is that shigatoxin-producing strains of E. coli are highly virulent and associated with 4,446 cases, 652 47 hospital admissions and 13 deaths. The authors should correct the sentence accordingly.
Lines 48 to 50 – It is not correct to say that Listeria monocytogenes is the most severe disease. First, L. monocytogenes is not a disease, it is a bacterium. The disease it causes is called listeriosis. Secondly, listeriosis has a reported death rate of 30% and several other infectious diseases have much higher mortality rates (e.g., Ebola infections, Naegleria infections). The authors should correct to “one of the most severe food-borne diseases”.
Line 51 – Wordy sentence. Please rephrase. It would suffice to say: The increased resistance to antibiotics in these pathogens…
Line 71 – Do the authors mean “soil” (as in agricultural soil), rather than “ground”?
Line 75 – 80 – Why is it important/interesting to isolate potential probiotic bacteria from mice and apply them is these same animals? What potential further applications might these have? How could the findings of this study be further applied within the framework of the One Health concept? I don’t think the authors mean to give probiotics to mice as a means of preventing zoonoses, or to apply the strains they isolated in other animals/humans. Are they merely using mice as a model? How can their data from the mice model be useful to better understand the potential of probiotics as an aid in the control of food-borne diseases? Or their purpose is to highlight the potentialities of bacteriolytic proteins. This has to be clearly explained, otherwise the study will lack interest and attractivity to prospective readers. The authors need to rework their manuscript, in order to give a better insight into the interest and impact their research will have. This part of the manuscript (as well as the Discussion and Conclusions sections), need to be rewritten, to clarify these aspects.
Line 89 – Please replace the word “light” with the word “lumen”.
Line 135 – Please correct Lc. Lactis to Lc. lactis.
421 – 497 – Please revise the Discussion section, in line with the comments provided above.
Author Response
"Please see the attachment."

Reviewer 2 Report
Please attend to the observations included in the attached file

Author Response
"Please see the attachment."

Round 2
Reviewer 1 Report
The authors implemented all changes required in the form of the manuscript. However, the deeper changes were not carried out. The manuscript still lacks a proper link with the potential applications of this research. In my previous revision, I pointed out questions that needed addressing to clarify these issues and they have not been answered in the present version of the manuscript. The authors response was very superficial and did not address all of the raised issues. I recommend the authors go back to those questions and address them, otherwise the interest of the manuscript for the journal's readers will be affected.
Some minor English corrections are still needed. Examples:
- please capitalize the initial of the word recognized (line 70)
- please replace the word paperwork with literature (line 433).
Author Response
Author 1. Comments and Suggestions
Dear reviewer, we thank you for taking the time to read and suggest improvements to this manuscript. We have worked to best answer the questions asked to highlight the importance of this work and applied the suggested changes to the text, these changes are highlighted in another color for quick identification, which are described in greater detail below:
Introduction
How can your data be useful to better understand the potential of probiotics as an aid in the control of foodborne illness?
Line 60-66, 67-80
The use of antimicrobials at large scale has been documented in livestock production. In addition to causing resistant bacteria that, in turn, provoke infections in humans, it has been found that animals excrete a significant percentage (75%–90%) of non-metabolized antimicrobials, which are scattered in the environment. [4]. The increased resistance to antibiotics in these pathogens contributes to the severity of the diseases they provoke, as observed recently in human and animal isolations of multi-resistant Campylobacter spp. and E. coli strains [5, 6].
Among the key strategies to address the constant infectious illnesses and the emerging challenges, the One Health framework and the objective to achieve good health and well-being are considered. Both are essential multi-faceted frameworks to promote good health for everyone. Taking into account the importance of health for sustainable development, the 2030 Agenda highlights the complexity and interconnection of both schemes, and considering the emerging global health priorities as well, which are not explicitly included in the sustainable development goals, but require action, as it is the case of resiliency to antimicrobials [7]. In the development of alternatives to reduce the use of antimicrobials is the use of vaccines, phage therapy and probiotics [1,4]. Probiotics are defined as “alive microorganisms that, when administered in adequate quantities, have a beneficial impact on the health of the host [8]. Among the studied beneficial effects are the control of pathogens, immunologic effects such as the preservation of the intestinal barrier and immunomodulatory activity. Lactic Acid Bacteria (LAB) are the primary microorganisms used as probiotics. These are administered with the purpose of controlling pathogen microorganisms and improve the defense natural mechanisms of humans and animals. It has been suggested that the use of these strains in animal production can reduce the risk of pathogen transference from food to humans, and it can be an alternative to the use of antibiotic growth promoters, therefore reducing the development of bacterial resistance [9].
Why is it important/interesting to isolate potential probiotic bacteria from mice?
Line 103-105
Animal gastrointestinal tract represents the source with the highest potential to obtain LAB [16], mainly Lactobacillus spp. Specifically, huge proportions of Lactobacillus have been identified in experimental mice, as well as the importance of this genera in the control of pathogens and the prevention of gut dysbiosis [20, 21].
What potential further applications might these have?
Line 108-113
It is expected that the results of this study will impact the detection of new strains and their metabolites with antibacterial activity in the search of strategies for the inhibition of these pathogens of clinical and zoonotic interest. Furthermore, the presence of enzymes with broad-spectrum bacteriolytic activity can be safely applied as promoters of animal health growth as an alternative to the use of antibiotics for the control of antibiotic resistant pathogenic bacteria.
In the Discussion and Conclusions sections, we rewritten to give a better idea of the interest and impact of the research.
Discussion
How could the findings of this study be further applied within the framework of the One Health concept?
Line 527-547
The findings of this study can be applied from the perspective of the sustainable health and well-being objective [7] and the multidisciplinary effort with the One Health framework to reduce health risks such as sanitary problems, zoonotic illnesses and antimicrobial resiliency, because these issues represent a growing threat for human and animal well-being. In the search of alternative solutions, there is the use of new probiotic strains and their bioactive components to reduce the use of antimicrobials because they provide health benefits and they do not generate risks of resistance mechanisms. In this context, our research is important because it explores a new niche in the pursuit of Lactobacillus strains, which displayed inhibitory capacity against test pathogen microorganisms; C. jejuni, which is considered the main bacterial cause of gastroenteritis in humans around the world and the zoonotic pathogen responsible for gastroenteritis in humans associated to chicken meat consumption [52]. E. coli REPEC, identified as one of the main causes of diarrhea in humans and animals [2], and acknowledged as the primary bacteria that causes diarrhea in children under five years old [53] and L. monocytogenes, considered as one emergent Foodborne Pathogen, provokes severe health issues that appear sporadically or as an epidemic [54]. Due to the inhibitory capacity of the 16 Ligilactobacillus murinus isolated strains presented in this study, as well as the identification of their antibacterial metabolites, a feasible alternative is suggested for the prevention and control of these pathogens in order to promote animal health and well-being and, subsequently, human health as well.
Conclusions
Line 549-555
The molecular identification of 16 strains of Ligilactobacillus murinus isolated from mice intestinal tract, its wide spectrum antibacterial activity against clinical and zoonotic pathogens and the presence of bacteriolytic enzymes, one of them being PGH, are an important finding that contributes to the antagonist potential of these strains. In addition, these enzymes have a potential application as alternative antibacterial agents and treatment of infections caused by antibiotic resistant bacteria, because the molecular mechanisms of resistance to antibiotics are not related to the hydrolysis of the PG.